# The Prognostic Utilities of Various Risk Factors for Laryngeal Squamous Cell Carcinoma: A Systematic Review and Meta-Analysis

**DOI:** 10.3390/medicina59030497

**Published:** 2023-03-02

**Authors:** Do Hyun Kim, Sung Won Kim, Jae Sang Han, Geun-Jeon Kim, Mohammed Abdullah Basurrah, Se Hwan Hwang

**Affiliations:** 1Department of Otolaryngology-Head and Neck Surgery, Seoul Saint Mary’s Hospital, College of Medicine, The Catholic University of Korea, 222 Banpo-daero, Seocho-gu, Seoul 06591, Republic of Korea; 2Department of Surgery, College of Medicine, Taif University, Taif 21944, Saudi Arabia; 3Department of Otolaryngology-Head and Neck Surgery, Bucheon Saint Mary’s Hospital, College of Medicine, The Catholic University of Korea, 222 Banpo-daero, Seocho-gu, Seoul 06591, Republic of Korea

**Keywords:** prognosis, risk factors, survival rate, laryngeal neoplasms, meta-analysis

## Abstract

*Objective*: To assess the prognostic utilities of various risk factors for laryngeal squamous cell carcinoma. *Methods*: Six databases were searched to January 2022. Hazard ratios for overall survival and disease-free survival were collected and study characteristics were recorded. The risk of bias was evaluated using the Newcastle–Ottawa scale. *Results*: Twenty-eight studies involving 32,128 patients were finally included. In terms of overall survival, older age, a history of alcohol consumption, a high Charlson comorbidity index score, a high TNM stage (III and IV), a high tumor stage (III and IV), nodal involvement, poor pathological differentiation, primary chemoradiotherapy and radiotherapy were associated with increased risks of death. In terms of disease-free survival, older age (≥60 years), TNM stages III and IV, tumor stages III and IV, supraglottic tumors, and nodal involvement all increased the risk of death. *Conclusions*: The TNM stage importantly predicts overall survival, and tumor location predicts the disease-free survival of patients with laryngeal squamous cell carcinoma. Of patients with risk factors, the Charlson comorbidity index usefully predicts overall survival.

## 1. Introduction

Laryngeal cancer is one of the most common cancers of the head and neck, associated with significant morbidity and mortality [1,2]. Internationally, approximately 1,700,000 cases of laryngeal squamous cell carcinoma (LSCC) are reported each year, and almost 90,000 patients die [3]. Despite significant advances in instruments such as flexible laryngoscopes, surgical methods, and chemoradiation therapy, the mortality rate remains high; the 5-year survival rate is 64% because about two-thirds of patients have advanced cancer at the time of diagnosis, rendering prognosis poor [1]. Additionally, despite improvements in treatment modalities, the American Cancer Society reported that the 5-year survival rate for patients with laryngeal cancer tends to decrease [4]. Advanced laryngeal cancer can be treated with radiation therapy alone, combination chemotherapy, or combination chemotherapy and radiation therapy with total laryngectomy [5]. Recently, it has been reported that the combination of radiation and chemotherapy provides organ preservation with equivalent survival rates compared to conventional treatment using surgical resection and adjuvant radiation therapy [6]. Treatment decisions are based on tumor staging according to guidelines such as the American Joint Committee on Cancer TNM classification for laryngeal function, the patient’s general condition, and patient and physician preference.

Patient-related factors including age, sex, smoking status, alcohol consumption, and comorbidities, as well as tumor-related factors such as TNM status, tumor location, pathological differentiation, and treatment, influence the prognosis. An understanding of prognostic risk factors would improve both treatment and survival. However, it is important to evaluate which factors are more important among various risk factors. However, such studies are insufficient [3]. Additionally, to the best of our knowledge, no meta-analysis has yet evaluated the impacts of individual risk factors (including comorbidities) on LSCC prognosis. Therefore, we meta-analyzed the effects of various risk factors on prognosis. This affords important insights that can improve treatments [2]. Through this study, it would be possible to improve the understanding of which factors among the risk factors of laryngeal cancer have a greater impact on the patient’s overall survival and disease-free survival.

## 2. Materials and Methods

### 2.1. Study Registration

We used the optimal surgical literature search method [7], and reported the results as suggested by the MOOSE (Meta-analysis of Observational Studies in Epidemiology) guideline [8]. The study protocol was prospectively registered on the Open Science Framework (https://osf.io/6zrcp/ accessed on 10 October 2022).

### 2.2. Literature Search

PubMed, Embase, the Web of Science, SCOPUS, Google Scholar, and the Cochrane Central Register of Controlled Trials were searched from the inception of publication of the relevant study to January 2023. The key search terms were laryngeal carcinoma, laryngeal neoplasm, larynx neoplasms, neoplasm, larynx, laryngeal cancer, risk factors, prognosis, survival, hazard ratio, overall survival (OS) rate, and disease-free survival (DFS). A librarian with more than 10 years experience searched all listed references, and the authors complemented the keyword-based searches by the combinations of all possible keywords with hand screening of references listed in the retrieved articles. Two independent reviewers with more than 5 years experience selected the studies. All references listed were also searched. Two independent reviewers selected studies on LSCC prognostic factors and survival rates via title, abstract, and text review.

### 2.3. Selection Criteria

The inclusion criteria were: (1) a human study; (2) exploration of the relationships between various risk factors and LSCC prognosis; and (3) survival data and prognostic information including hazard ratios (HRs) with 95% confidence intervals (CIs) in terms of OS or DFS after surgery, radiation therapy, chemotherapy, or combination therapy. The exclusion criteria were: (1) case reports, reviews, book chapters, books, editorial letters, opinion papers, or animal studies; (2) double primary cancers, metastatic cancers, or invasion of adjacent cancers (e.g., esophageal cancer); (3) any history of another head and neck cancer such as tongue or salivary gland cancer; (4) studies not in English and, (5) the lack of adequate prognostic data. A flow diagram of study selection is shown in Figure 1.

### 2.4. Data Curation and Methodological Assessment

Data were independently extracted in an agreed form by two reviewers [9,10,11], who also evaluated the risk of bias. Any differences of opinion were resolved in a panel discussion with a third reviewer. We collected the name of the lead author, the year of publication, the study design, patient numbers, age, sex, nationality, survival outcomes, TNM stages (the American Joint Committee on Cancer TNM system), and HRs with 95% CIs for OS and DFS [2,6,12,13,14,15,16,17,18,19,20,21,22,23,24,25,26,27,28,29,30,31,32]. HRs and 95% CIs were assessed using the usual methods if not specifically indicated in any study [33,34]. We subjected the OSs, the HRs, and the 95% CIs to multivariate analyses [35]. Statistical significance was determined based on the *p* values. The methodological quality (risk of bias) of each study was assessed using the Newcastle–Ottawa scale. The scores range from 0 to 9; a score ≥ 6 indicates high quality [36]. Data reported only in graphical plots were not extracted for pooled meta-analysis unless specific numeral points were discernible or the authors of the relevant studies were able to verify the data. In the event of missing or incomplete data, attempts were made to request data directly from the authors.

### 2.5. Statistical Analysis and Outcome Measurements

The meta-analysis was performed using R software (the R Foundation, Vienna, Austria). Heterogeneity was assessed by employing the Q statistic. The extent of heterogeneity was measured using the I^2^ method. An I^2^ of 75 to 100% indicated high, 50 to 75% medium, and 25 to 50% low heterogeneity. An I^2^ value < 25% reflected non-heterogeneity. When I^2^ < 50%, a fixed-effects model was used; when I^2^ ≥ 50%, a random-effects model was applied. Subgroup analyses were performed according to the TNM stage [all stages (I–IV) or advanced stages (III–IV)]. Publication bias was assessed using the Egger linear regression test and by drawing Begg funnel plots. Sensitivity analysis was achieved by measuring how the removal of single studies modified the total effects.

## 3. Results

### 3.1. Study Selection

Twenty-eight studies involving 32,128 patients were included in total (Table 1). Search terms and queries were presented as Appendix A. The risks of bias are listed in Appendix A. In terms of potential publication bias, the Egger test result was significant (*p* > 0.05), suggesting a source of such bias that was not evident in the studies. Begg funnel plots by sex (*p* = 0.06637) and tumor stage (*p* = 0.0501) revealed no publication bias (Appendix AA,B). However, funnel plot analyses by age (*p* < 0.0001) and nodal involvement (*p* < 0.0001) suggested a source of bias (Appendix AC,D). We therefore performed the Duval and Tweedie trim-and-fill test and found no significant difference between the observed and adjusted values (age 1.2217, *p* < 0.0001 vs. 1.0599; *p* = 0.0454; nodal involvement 2.0537, *p* < 0.0001 vs. 1.3578; *p* = 0.0048). Therefore, we concluded that the studies were not biased and that the results reliably reflected the clinical features. However, given the small numbers of relevant studies (<10), we did not use the Egger linear regression test or draw Begg funnel plots for alcohol consumption, smoking status, the Charlson comorbidity index, clinical stage, pathological differentiation, or tumor subsite.

### 3.2. Overall Survival by Patient-, Tumor-, and Treatment-Related Factors

Patient-related factors included age (≥60 or <60 years), sex, current smoking (yes or no), current alcohol consumption (yes or no), and the Charlson comorbidity index (≥2, 1, or 0). The tumor-related factors were the TNM stage (III and IV vs. I and II); tumor stage (III and IV vs. I and II); tumor location (subglottis, supraglottis, or glottis); nodal involvement (yes or no); and the extent of pathological differentiation (poor, moderate, or high). The treatment-related factors were primary chemoradiotherapy, radiotherapy, or total laryngectomy (Table 2).

In terms of overall survival, older age (>60 years) [hazard ratio (HR) 1.1300, 95% confidence interval (CI) [1.0908; 1.1705]; *p* < 0.0001]; a smoking history (HR 1.2926, 95% CI (1.0999; 1.5191); *p* = 0.0018); a history of alcohol consumption [HR 1.1979, 95% CI (1.0696; 1.3415); *p* = 0.0018]; a high Charlson comorbidity index score [≥2 vs. 0; HR 1.6716, 95% CI (1.3533; 2.0647); *p* < 0.0001 and 1 vs. 0; HR 1.3153, 95% CI (1.2299; 1.4068); *p* < 0.0001]; a high TNM stage (III and IV) [HR 2.4583, 95% CI (1.8323; 3.2980); *p* < 0.0001); a high tumor stage (III and IV) [HR 1.5648, 95% CI (1.2363; 1.9806); *p* < 0.0001); nodal involvement [HR 1.9439, 95% CI (1.6235; 2.3276); *p* < 0.0001], poor pathological differentiation [moderate vs. high; HR 1.2820, 95% CI (1.0659; 1.5419); *p* < 0.0001 and poor vs. high (HR 1.6951, 95% CI [1.5394; 1.8665]; *p* < 0.0001); supraglottic tumor (vs. glottis tumor) (HR 1.3740, 95% CI [1.0730; 1.7594; *p* = 0.0118); primary chemoradiotherapy and radiotherapy (vs. primary chemoradiotherapy and total laryngectomy [HR 1.4004, 95% CI (1.1639; 1.6850); *p* = 0.0004); and primary radiotherapy vs. total laryngectomy [HR 1.5418, 95% CI (1.1531; 2.0616); *p* = 0.0035] were associated with increased risk of death (Figure 2). In terms of disease-free survival, those of older age (≥60 years), TNM stages III and IV, tumor stages III and IV, with supraglottic tumors, and nodal involvement, were at increased risk of death compared to younger patients (<60 years) [HR 1.0070, 95% CI (1.0005; 1.0136); *p* = 0.0359), as were those of TNM stages I and II [HR 2.3987, 95% CI (2.0956; 2.7456); *p* < 0.0001), those of tumor stages I and II [HR 1.8441, 95% CI (1.4507; 2.3441), *p* < 0.0001), patients with glottis tumors [HR 1.6371, 95% CI (1.0162; 2.6373); *p* < 0.0001), and those lacking nodal involvement [HR 1.6174, 95% CI (1.4560; 1.7968); *p* < 0.0001).

### 3.3. Subgroups Analysis

The prognostic values revealed significant heterogeneity because patients of different TNM stages (all stages; I–IV) or advanced stages only (III–IV) were included. Therefore, we performed subgroup analyses by the TNM stage. In terms of OS, significant heterogeneity was apparent in the outcomes by age, the Charlson comorbidity index score, the TNM stage, nodal involvement, sex, tumor location (subglottis, supraglottis, or glottis), and treatment modality (primary chemoradiotherapy, radiotherapy, or total laryngectomy). Only one study compared subgroups in terms of nodal involvement, tumor location, and treatment modality; therefore, we could not perform subgroup comparisons. The OSs of subgroups classified by age, the Charlson comorbidity index score, and sex did not differ significantly, which may mean that these factors exert similar effects regardless of TNM stage. However, tumor location (supraglottis vs. glottis) exhibited significant heterogeneity. For patients of all disease stages, subgroup analysis revealed that supraglottic lesions generally reduced survival compared to that of subjects with glottis lesions, but not in patients of advanced stage.

### 3.4. Sensitivity Analyses

Sensitivity analyses were performed to evaluate whether the pooled estimates of overall or disease-free survival by patient-, tumor-, and treatment-related factors were different by omitting a different study each time and repeating the meta-analyses. Finally, the results were all consistent with the above outcomes.

## 4. Discussion

We found that older age, a higher Charlson comorbidity index (≥1), advanced TMN stage (III and IV), advanced tumor stage, nodal involvement, pathological differentiation, primary chemoradiotherapy and radiotherapy were all associated with poorer OSs. Similarly, older age, advanced TMN stage (III and IV), advanced tumor stage, a supraglottic lesion, and nodal involvement were associated with poorer DFS. For LSCC, in common with many other tumors, tumor stage (T stage), cervical lymph node metastasis, and the clinical stage are key predictors of prognosis [19,25,28]. Additionally, we found that the T stage, lymph node metastasis, and the clinical stage significantly affected prognosis; the clinical stage exerted a greater influence on survival than did the T stage or lymph node metastasis, probably because clinical staging comprehensively reflects the tumor stage and nodal invasion and metastasis [2]. In LSCC patients, apart from the clinical stage, tumor location is significantly prognostic. A supraglottic laryngeal cancer can grow significantly before the development of symptoms. Given the abundant lymphatic drainage, nodal metastases are often present at the time of onset of such symptoms [37]. In contrast, in those with glottis cancer, lymphatic drainage around the glottis is poor, and early lymph node metastases are rare (<5% of patients). Therefore, it could be predicted that early supraglottis cancers would be associated with poorer prognoses than early glottis cancers [2]. However, many advanced glottis cancers in fact arise in the laryngeal ventricle, and easily spread to the supraglottis and paraglottic space [37]. Therefore, in our subgroup analyses, the survival rates of those with advanced glottis and supraglottis cancers were similar, but when all stages (including early stages) were included, supraglottis cancers evidenced a poorer prognosis. We also found that the tumor pathological differentiation status was prognostic, as is true of many tumors [28,29]. Progression of dedifferentiation (to moderate and poor) was associated with significantly poorer prognosis, consistent with previous reports.

The treatment modality can also significantly Impact prognosis. Several early studies reported similar survival rates when chemoradiotherapy (rather than surgery) was used to treat advanced-stage laryngeal cancer (it was sought to preserve laryngeal function) [38,39]. Therefore, after publication of these prospective randomized trials, the use of primary chemoradiation therapy for LSCC increased significantly. However, after the papers were published, clinical studies showed that primary chemoradiotherapy was in fact not as good as primary surgery [32,40,41]. Dziegielewski et al. found that the OS of patients with advanced laryngeal cancers (stages T3 and T4) increased when they were treated with concurrent surgery and radiotherapy or chemoradiotherapy, compared to radiotherapy or chemoradiotherapy alone [13]. We similarly found that primary total laryngectomy was associated with better survival than radiotherapy or chemoradiotherapy alone.

Although prognostic tumor- and treatment-related factors of LSCC patients have been studied, patient-related factors are often overlooked; they are nonetheless important. Comorbidities affect disease prognosis, treatment choices, and outcomes. The Charlson comorbidity index (developed in 1987) is today widely used to identify comorbidities and to apply weighted or pathophysiological severity ratings [39]. The index validity predicts outcomes and mortality risks, including those of head and neck cancer patients [40]. Several studies found that comorbidities significantly influenced LSCC patient survival [2,17,21,22,24]. We found that the index usefully predicted overall survival, as have previous reports [42,43]. The presence of significant comorbidities is common in head and neck cancer, with approximately 30–50% of patients having at least one comorbidity [44]. With the accumulation of evidence, management of comorbidities and chronic diseases in the survival stage of head and neck cancer management is becoming increasingly important [45]. This is reported to be due to higher non-cancer related mortality, which can affect about 10–30% of patients [44]. It is also reported that about 20–30% of patients need to modify treatment decisions due to comorbidity [46]. In this context, the Charlson comorbidity index score was shown to be the next strongest risk factor for poor overall survival following the TNM clinical stage. Age may affect LSCC prognosis. On univariate analysis, Wong et al. showed that age was not associated with poor survival in patients with newly diagnosed laryngeal cancer [16]. However, Graboyes et al. found that age was prognostic with univariate and multivariate analyses of advanced laryngeal cancer [21]. Our results are consistent with those of Graboyes et al.; the survival rate of older patients was lower than that of younger subjects. Aging is associated with decreased organ function, more comorbidities, and cognitive dysfunction that adversely affect the immune system, the responses to treatment, and prognosis [47]. In addition, older patients generally receive less intensive care than younger patients [1]; they tend to be prescribed RT or CRT more often than total laryngectomy [13]. This may reduce the DFS of older patients. Smoking and alcohol consumption were associated with poor prognosis. Toxins in tobacco smoke and alcohol metabolites impair the innate defenses, adversely modulate antigen presentation, and trigger chronic inflammation of mucosal surfaces. In the case of alcohol, it can cause gastroesophageal reflux, increasing the chance of contact between the upper aerodigestive epithelium and dietary carcinogens. This can cause transformation of epithelial lesions and tumorigenesis [48]. These materials also bind to the DNA of mucosal cells, triggering mutations and malignant transformation [29]. Infiltration of immune cells from LSCC, such as mast cells, neutrophils, and macrophages, can promote cancer development and tumor angiogenesis, and produce small molecules, including cytokines, chemokines, and growth factors, which allow tumors to avoid host immune responses [49,50]. Such genetic and molecular alterations constitute a distinct pathological entity causing poor prognosis [41]. Field cancerization may be prognostically important in such patients [41,50]. Recently, studies on the role of genetic mutations in the etiology of laryngeal cancer have also been conducted. This may lead to a better understanding of mechanisms and the discovery of effective molecular markers for the development of novel screening and treatment strategies [51]. However, more investigation is required. Through this study, we reviewed LSCC risk factors and were able to identify which factors could have a greater impact on overall survival and disease-free survival of patients. This information can be important to guide consideration of adjuvant treatment modalities and preoperative discussions about treatment goals.

Our meta-analysis had several limitations. First, as some data evidenced significant heterogeneity, we used a random-effects model to perform subgroup analyses. Second, the institutional setting and (probably) unknown factors affect the prognostic utilities of various OS and DFS risk factors for OS and DFS. Additionally, it is difficult to analyze by setting the conditions of the patients in the included individual studies in the same way. Although this is a limitation of meta-analysis, it has the advantage of generalizing the results of independent studies with differences in the experimental environment. Third, 9 of the 23 articles were from China; selection bias may be in play because regional grouping was not considered. Additionally, most patients were male, which may reflect the nature of the disease per se, but may also reflect a gender bias [3]. These observations may explain the heterogeneity of our prognostic outcomes. Fourth, any cross-sectional work may over- or under-estimate disease prevalence. Finally, we encountered methodological heterogeneity and inadequate reporting of methods. To overcome these limitations, large-scale homogenous population studies are required. Fifth, in our study, we included the only studies written in English, which could restrict the diversity of results. However, exclusion of non-English publications from systematic reviews on clinical interventions had a minimal effect on overall conclusions and could be a viable methodological shortcut [52].

## 5. Conclusions

This meta-analysis found that TNM stage significantly predicts overall survival and tumor location predicts disease-free survival in laryngeal squamous cell carcinoma patients. In patients with risk factors, the Charlson comorbidity index is a useful predictor of overall survival. All tumor-, treatment-, and patient-related factors affect the prognosis of LSCC patients to varying extents. Many risk factors (including comorbidities) are often overlooked during prognostic evaluation. However, they are as important as other factors.

## Figures and Tables

**Figure 1 medicina-59-00497-f001:**
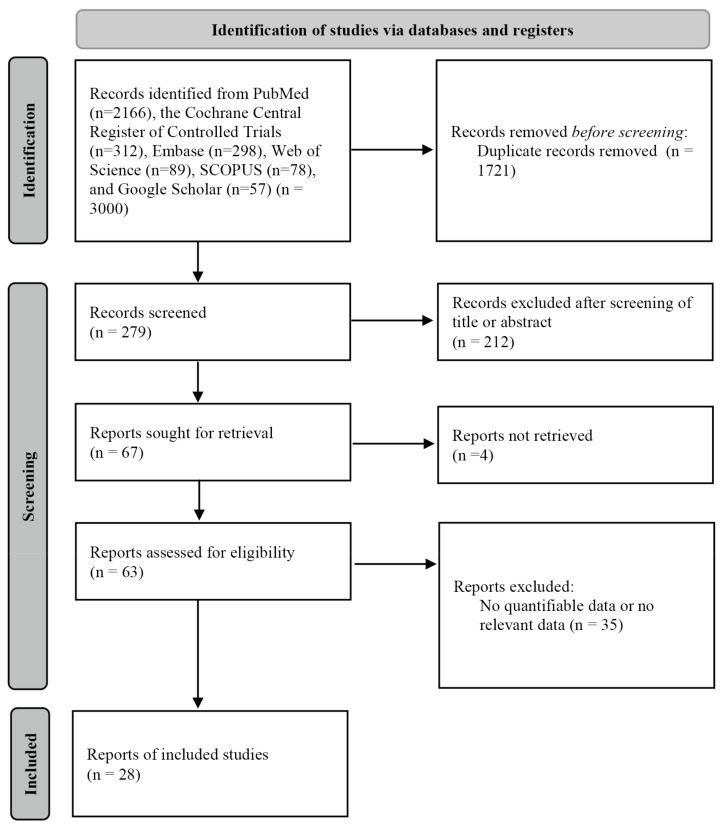
Study selection.

**Figure 2 medicina-59-00497-f002:**
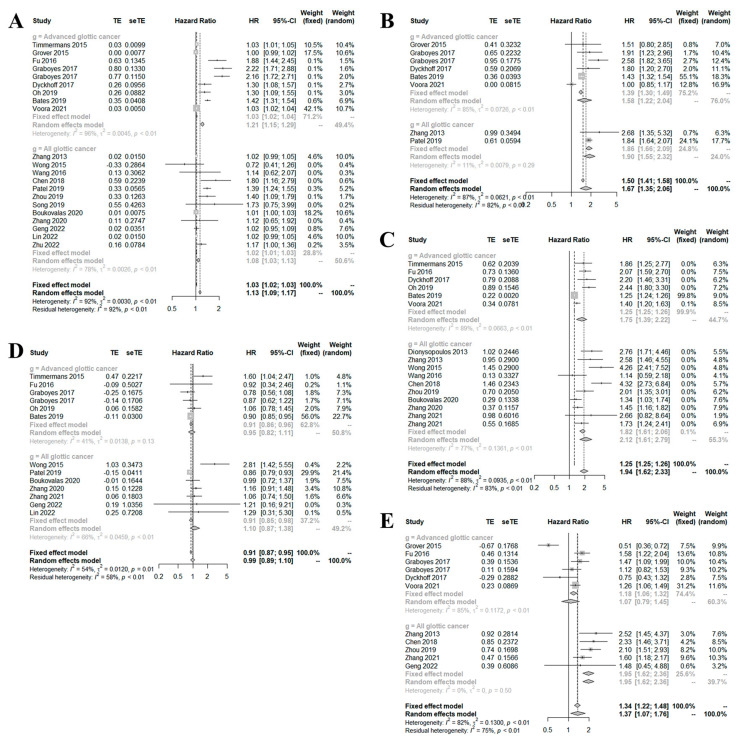
Forest plots of the effects of age (**A**), the Charlson comorbidity index score (**B**), nodal involvement (**C**), sex (**D**), and tumor location (**E**) on overall survival.

**Table 1 medicina-59-00497-t001:** The characteristics of the included studies.

Study	Design	Number	Age, Median (Range) or Mean (SD), y	Sex (Male/Female)	Nation	Treatment	Tumor Subsite (Supraglottic/Glottic/ Subglottic)	Laryngeal Tumor Stage	Extracted Outcomes
Nichols 2012 [12]	Cohort	75	NA	66/9	UK	Radiotherapy	Early glottis cancer	T1/T2	Age, gender, T stage, smoking, alcohol
Dziegielewski 2012 [13]	Cohort	258	64.2 (36–92)	205/53	Canada	TL with adjuvant RT ± CT (TL-R/CT), RT, and chemotherapy–radiotherapy (CRT)	Supraglottic/glottic/subglottic	T3/T4a	Treatment modality
Dionysopoulos 2013 [14]	Cohort	289	63 (36–82)	277/12	Greece	Chordectomy, TL ± postoperative radiation	Supraglottic/glottic/subglottic/transglottic	T1/T2/T3/T4	Nodal involvement, tumor subsite
Zhang 2013 [2]	Cohort	205	61.8 ± 10.6	197/8	China	Total laryngectomy, partial laryngectomy, or CO2 laser surgery ± postoperative radiation or CT	Supraglottic/glottic	T1/T2/T3/T4	Age, TMN stage, T stage, node involvement, tumor subsite, smoking, alcohol, Charlson score, patholic differentiation
Timmermans 2015 [15]	Cohort	166	61.9 (11.3)	124/42	The Netherlands	Radiotherapy, chemoradiotherapy, or total laryngectomy with postoperative radiotherapy	Supraglottic/glottic/subglottic/transglottic	T3/T4	Treatment modality, Age, sex, T stage, node involvement
Wong 2015 [16]	Cohort	140	66 (36–92)	121/19	UK	Primary surgery, surgery with adjuvant chemoradiotherapy or radiotherapy, radical radiotherapy, and chemoradiotherapy	Not commented	T1/T2/T3/T4	Sex, age, smoking, node involvement, TMN stage
Grover 2015 [17]	Cohort	969	59.2 (10.4)	774/195	USA	Total laryngectomy (TL) plus adjuvant therapy and larynx preservation chemoradiation (LP-CRT)	Supraglottic/glottic/subglottic/transglottic	T4a	Age, TMN stage, node involvement, Charlson score, tumor subsite
Wang 2016 [18]	Cohort	120	60.6 ± 8.6	118/2	China	Total laryngectomy, partial laryngectomy, or CO2 laser surgery plus postoperative radiation ± CT	Supraglottic/glottic/subglottic	T1/T2/T3/T4	Age, smoking, alcohol, tumor subsite, T stage, node involvement, TMN stage, pathologic differentiation
Tu 2015 [19]	Cohort	141	59 (36–87)	137/4	China	Total laryngectomy, partial laryngectomy, or CO_2_ laser surgery	Supraglottic/glottic/subglottic	T1/T2/T3/T4	T stage, node involvement
Fu 2016 [20]	Cohort	420	60 ± 9.1 (33–84)	413/7	China	Total laryngectomy (TL) ± adjuvant therapy	Supraglottic/glottic/subglottic	T3/T4	Age, sex, smoking, alcohol, tumor subsite, T stage, node involvement, TMN stage, pathologic differentiation
Graboyes 2017 [21]	Cohort	1460	NA	531/143	USA	Partial laryngectomy or Total laryngectomy ± postoperative radiation	Supraglottic/glottic/subglottic/transglottic	T3	Age, sex, Charlson score, tumor subsite, pathologic differentiation
Dyckhoff 2017 [22]	Cohort	769	61.9 (9.7)	626/58	Germany	Primary chemo-radiotherapy (CRT) or primary radiotherapy alone (RT), total laryngectomy followed by adjuvant (chemo)radiotherapy	Supraglottic/glottic/subglottic/transglottic	T4	Treatment modality, age, node involvement, tumor subsite, Charlson score
Birkeland 2017 [23]	Cohort	244	NA	208/36	USA	Total laryngectomy followed by adjuvant (chemo)radiotherapy	Supraglottic/glottic	T1/T2/T3/T4	T stage
Cheraghlo 2018 [24]	Cohort	726	NA	528/198	USA	Total laryngectomy, open partial laryngectomy, and endoscopic partial laryngectomy	Supraglottic/glottic/subglottic/transglottic	T1/T2	Age, sex, Charlson score, tumor subsite, T stage, node involvement
Chen 2018 [25]	Cohort	361	60 (35–87)	353/8	China	Total or partial laryngectomy without neoadjuvant chemotherapy or radiotherapy	Supraglottic/glottic/subglottic	T1/T2/T3/T4	Age, sex, tumor subsite, T stage, node involvement, TMN stage, pathologic differentiation
Oh 2019 [26]	Cohort	329	62 (57–66)	30/6	Canada	Surgery alone, surgery with adjuvant radiotherapy (Sx/RT), radiation alone (RT), and radiation with concurrent chemoradiotherapy (chemoRT)	Not described	T4a	Treatment modality, age, alcohol, sex, node involvement
Patel 2019 [27]	Cohort	8703	NA	6601/2102	USA	Chemoradiation (CRT) or partial laryngectomy (PL) and total laryngectomy (TL) with or without adjuvant therapy	Supraglottic/glottic	T2/T3/T4	Age, sex, Charlson score, pathologic differentiation, T stage, node involvement, treatment modality
Zhou 2019 [28]	Cohort	232	63 (39–81)	192/40	China	Partial or total laryngectomy (±neck dissection) and postoperative radio-/chemotherapy	Supraglottic/glottic/subglottic	T1/T2/T3/T4	Age sex, smoking, alcohol, tumor subsite, T stage, node involvement, TMN stage, pathologic differentiation
Song 2019 [29]	Cohort	137	NA	133/4	China	Total laryngectomy	Supraglottic/glottic/subglottic/transglottic	T1/T2/T3/T4	Tumor subsite, age, pathologic differentiation
Bates 2019 [32]	Cohort	11,237	NA	8472/3538	USA	Chemoradiotherapy (cRT) and total laryngectomy (TL) with adjuvant RT	Not described	T3/T4	Age, sex, Charlson score, T stage, node involvement, treatment modality
Boukovalas 2020 [6]	Cohort	362	64	294/68	USA	Total laryngectomy	Not described	T1/T2/T3/T4	Age, sex, T stage, node involvement, smoking, alcohol
Zhang 2020 [30]	Cohort	207	NA	198/9	China	Partial or total laryngectomy	Supraglottic/glottic/subglottic/transglottic	T1/T2/T3/T4	Age, sex, alcohol, smoking, pathologic differentiation, TMN stage, T stage, node involvement
Lin 2021 [31]	Cohort	2094	NA	1712/382	China	Total laryngectomy	Supraglottic/glottic	T1/T2/T3/T4	Age, sex, pathologic differentiation, node involvement
Voora 2021	Cohort	1043	62.29 (8.13)	1039/4	USA	Chemoradiotherapy (cRT) and total laryngectomy (TL) with adjuvant RT	Supraglottic/glottic/subglottic/transglottic	T4a	Age, sex, Charlson score, alcohol, tumor subsite, smoking, node involvement, treatment modality
Zhang 2021	Cohort	211	62.19 (8.328)	164/47	China	Total or partial laryngectomy ± Adjuvant radiotherapy	Supraglottic/glottic/subglottic/transglottic	T1/T2/T3/T4	Sex, alcohol, tumor subsite, smoking, node involvement, TMN stage, pathologic differentiation
Geng 2022	Cohort	78	58.1 (51.1–62.1)	70/8	USA	Total or partial laryngectomy ± Adjuvant radiotherapy	Supraglottic/glottic/subglottic	T1/T2/T3/T4	Age, sex, tumor subsite, smoking, TMN stage, pathologic differentiation
Lin 2022 [31]	Cohort	154	60.90 (9.79)	147/7	China	Total or partial laryngectomy ± Adjuvant radiotherapy	Supraglottic/glottic/subglottic	T1/T2/T3/T4	Age, sex, alcohol, smoking, TMN stage, treatment modality
Zhu 2022	Cohort	998	56–70	946/52	China	Chemoradiotherapy (cRT) and total laryngectomy (TL) with adjuvant RT	Supraglottic/glottic/subglottic	T1/T2/T3/T4	Age, sex, tumor subsite, TMN stage, pathologic differentiation

CRT; chemoradiation therapy, TL; total laryngectomy, RT; radiation therapy, HR; Hazard ratio.

**Table 2 medicina-59-00497-t002:** Predictive values of various risk factors.

	Overall Survival	Disease-Free Survival
**Age (≥60 vs. 60)**	n = 21	n = 9
	1.1300 [1.0908; 1.1705]; *p* < 0.0001; I^2^ = 91.5%	1.0070 [1.0005; 1.0136]; *p* = 0.0359; I^2^ = 15.9%
**Sex (male vs. female)**	n = 13	n = 8
	0.9856 [0.8866; 1.0956]; *p* = 0.7883; I^2^ = 53.9%	1.1047 [0.7234; 1.6869]; *p* = 0.6448; I^2^ = 84.7%
**Smoking (yes vs. no)**	n = 11	n = 9
	1.2926 [1.0999; 1.5191]; *p* = 0.0018; I^2^ = 13.6%	1.2237 [0.9206; 1.6267]; *p* = 0.1644; I^2^ = 61.2%
**Alcohol (yes vs. no)**	n = 9	n = 7
	1.1979 [1.0696; 1.3415]; *p* = 0.0018; I^2^ = 33.1%	1.1861 [0.9588; 1.4673]; *p* = 0.1159; I^2^ = 54.1%
**Charlson score (≥2 vs. 0 or 1)**	n = 8	
	1.6716 [1.3533; 2.0647]; *p* < 0.0001; I^2^ = 86.6%	
**Charlson score (1 vs. 0)**	n = 5	
	1.3153 [1.2299; 1.4068]; *p* < 0.0001; I^2^ = 12.1%	
**TNM stage (III and IV vs. I and II)**	n = 9	n = 10
	2.4583 [1.8323; 3.2980]; *p* < 0.0001; I^2^ = 81.9%	2.3987 [2.0956; 2.7456]; *p* < 0.0001; I^2^ = 29.8%
**Tumor stage (III and IV vs. I and II)**	n = 10	n = 5
	1.5648 [1.2363; 1.9806]; *p* < 0.0001; I^2^ = 86.4%	1.8441 [1.4507; 2.3441]; *p*< 0.0001; I^2^ = 0.0%
**Tumor location (subglottis vs. glottis)**	n = 4	
	1.3956 [0.5841; 3.3341]; *p* = 0.4532; I^2^ = 84.1%	
**Tumor location (supraglottic vs. glottis)**	n = 11	n = 4
	1.3740 [1.0730; 1.7594]; *p* = 0.0118; I^2^ = 82.3%	1.6371 [1.0162; 2.6373]; *p* < 0.0001; I^2^ = 90.2%
**Tumor location (transglottic vs. glottis)**	n = 5	
	1.3699 [0.9348; 2.0076]; *p* = 0.1065; I^2^ = 69.3%	
**Node involvement (yes vs. no)**	n = 16	n = 10
	1.9439 [1.6235; 2.3276]; *p* < 0.0001; I^2^ = 87.6%	1.6174 [1.4560; 1.7968]; *p* < 0.0001; I^2^ = 46.6%
**Pathologic differentiation (moderate vs. high)**	n = 6	
	1.2820 [1.0659; 1.5419]; *p* < 0.0001; I^2^ = 51.3%	
**Pathologic differentiation (poor vs. high)**	n = 9	n = 3
	1.6951 [1.5394; 1.8665]; *p* < 0.0001; I^2^ = 43.2%	1.5336 [0.7875; 2.9864]; *p* = 0.2086; I^2^ = 90.1%
**Treatment modality (primary chemoradiotherapy vs. total laryngectomy)**	n = 10	
	1.4004 [1.1639; 1.6850]; *p* = 0.0004; I^2^ = 90.6%	
**Treatment modality (primary radiotherapy vs. total laryngectomy)**	n = 8	
	1.5418 [1.1531; 2.0616]; *p* = 0.0035; I^2^ = 91.6%	

## Data Availability

The raw data of individual articles used in this meta-analysis are included in the main text or supplementary data.

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
