# Peer review of "The Prognostic Utilities of Various Risk Factors for Laryngeal Squamous Cell Carcinoma: A Systematic Review and Meta-Analysis"

_medicina, 2023, doi:10.3390/medicina59030497_

Round 1
Reviewer 1 Report
With pleasure, I read the paper titled: “The prognostic utilities of various risk factors for laryngeal squamous cell carcinoma: A systematic review and meta-analysis”. Overall, the paper reads very well. The novelty lies in being the first meta-analysis to examine impact of risk factors on survival outcomes among patients with LSCC. The results are reported in depth which is a great strength of the report, and the summary tables/figures add beautification and crispy take-home-messages. I congratulate the authors on a well-done investigation. I have the following comments/suggestions below.
Introduction. Please enrich the introduction section with more background to highlight the gaps in literature. Please pinpoint how you research is going to fill up this literature gap. Please indicate if similar meta-analysis has been done previously. Please end the introduction section with a hypothesis.
Methods. Your research should be reported in line with MOOSE (Meta-analysis of Observational Studies in Epidemiology) since you included observational studies. Please provide the exact literature search used in all databases, or at least, you should provide the exact literature search for one database in Supplemental Table to check for reproducibility of your literature search. For Figure 1, you need to indicate how many citations were retrieved from each database. Please indicate statistical significance was determined based on what p value.
Results. Supplemental Figure 1 does not seem methodologically right, as funnel plots are not used for such purpose. Please consult the Cochrane Handbook. It should be removed. For Table 1, is this right: “Pathologic differentiation (moderate vs high)”, or it is supposed to be “Pathologic differentiation (high vs moderate)”. Please double-check. For Figure 2, in the legend, please provide the comparisons (for example, for age, it was male vs female) and etc. You may want to provide a Figure for DFS data like Figure 2. High between-study heterogeneity was significant for many variables and not only TNM stage. So, why did you decide to do subgroup analysis based on TNM stage only?
Discussion. Please briefly highlight the clinical implications and future research directions.
References. They are appropriate and up to date.
Language. The manuscript will benefit from minor polishing for English language.
Author Response
Introduction. Please enrich the introduction section with more background to highlight the gaps in literature. Please pinpoint how you research is going to fill up this literature gap. Please indicate if similar meta-analysis has been done previously. Please end the introduction section with a hypothesis.
â—Ž Reply:
We have added information about the background of starting this study in the introduction section. In addition, the need for research on which indicators should be considered more important among the risk factors of laryngeal cancer when making clinical judgments was further highlighted.
It was noted that no meta-analyses had previously been conducted on this topic. In addition, the introduction was completed with a hypothesis about the results to be obtained with this study.
Methods. Your research should be reported in line with MOOSE (Meta-analysis of Observational Studies in Epidemiology) since you included observational studies. Please provide the exact literature search used in all databases, or at least, you should provide the exact literature search for one database in Supplemental Table to check for reproducibility of your literature search. For Figure 1, you need to indicate how many citations were retrieved from each database. Please indicate statistical significance was determined based on what p value.
â—Ž Reply:
This manuscript was written in accordance with MOOSE (Meta-analysis of Observational Studies in Epidemiology) guideline, and a checklist was attached. References in the text have also been revised.
In addition, search terms and queries were added as Table S1.
We added a statement "statistical significance was determined based on what p" to the methods section.
Results. Supplemental Figure 1 does not seem methodologically right, as funnel plots are not used for such purpose. Please consult the Cochrane Handbook. It should be removed. For Table 1, is this right: “Pathologic differentiation (moderate vs high)”, or it is supposed to be “Pathologic differentiation (high vs moderate)”. Please double-check. For Figure 2, in the legend, please provide the comparisons (for example, for age, it was male vs female) and etc. You may want to provide a Figure for DFS data like Figure 2. High between-study heterogeneity was significant for many variables and not only TNM stage. So, why did you decide to do subgroup analysis based on TNM stage only?
â—Ž Reply:
Funnel plot is used to evaluate the publication bias of the outcomes. Figure S1 was Funnel plots. The sentences for Figure S1A, S1B was right for explanation of publication bias and funnel plot. However, the sentence for Figure S1C, S1D was not appropriate because these were for the Trim fill methods and results. Therefore, the location of Figure S1C, S1D was relocated in the results section.
As the comment of the reviewer, “Pathologic differentiation (high vs moderate) is right, so we corrected the typo and the comments of comparison were added in the Figure 2 legend.
For subgroups analysis, clear divisions between groups are necessary but unlike TNM stage, other many variables were not appropriate to conduct subgroup analysis.
Discussion. Please briefly highlight the clinical implications and future research directions.
â—Ž Reply:
We emphasized by adding clinical implications and future research directions to the discussion section.

Reviewer 2 Report
The authors present the systematic review and metaanalysis investigating the effects of various risk factors on prognosis in patients with larygenal carcinoma. They performed first meta-analysis about the impacts of individual risk factors on LSCC prognosis. The author team has insuffiently summarized the most recent and relevant papers on the subject. I would like to offer the following points for consideration by the authors towards the improvement of the manuscript:
1- Please explain the search strategy further. How did you perform systematic search using keyword (for example " laryngeal carcinoma " AND "prognosis" OR “risk factors”)
2- It is not clear exactly how the authors narrowed the number of articles. Please elaborate on the exclusion criteria (for example book chapters, books, case reports, editorial letters, review articles, retrospective studies, single-arm studies, and opinion papers; animal studies; studies not in English) with numbers.
3- I suggest improving the presentation and quality of Table S1 in terms of treatment, tumor subsite (supraglottic/glottic/subglottic)
4- Please explain why did not you include this study to analysis.
- https://doi.org/10.3389/fonc.2021.606010
5- Since there are new studies, it would be good to update the search range and repeat the analysis.
Author Response
1- Please explain the search strategy further. How did you perform systematic search using keyword (for example " laryngeal carcinoma " AND "prognosis" OR “risk factors”)
â—Ž Reply:
Search terms and queries were added as Table S1.
2- It is not clear exactly how the authors narrowed the number of articles. Please elaborate on the exclusion criteria (for example book chapters, books, case reports, editorial letters, review articles, retrospective studies, single-arm studies, and opinion papers; animal studies; studies not in English) with numbers.
â—Ž Reply:
We added to the text that we excluded book chapters, books, editorial letters, opinion papers, animal studies, and studies not in English.
3- I suggest improving the presentation and quality of Table S1 in terms of treatment, tumor subsite (supraglottic/glottic/subglottic)
â—Ž Reply:
We added the columns related to of treatment and tumor subsite in Table 1.
4- Please explain why did not you include this study to analysis.
- https://doi.org/10.3389/fonc.2021.606010
â—Ž Reply:
We added the commented study in the analysis of the revised version.
5- Since there are new studies, it would be good to update the search range and repeat the analysis.
â—Ž Reply:
We revised manuscript by searching PubMed, Embase, the Web of Science, SCOPUS, Google Scholar, and the Cochrane Central Register of Controlled Trials databases through January 2023. Five studies were added, and the results were updated accordingly.

Round 2
Reviewer 2 Report
I am satisfied that the authors have addressed all of my previous concerns about the article. It is now much improved and I feel that it is now suitable for publication.